# Distinction between Antimicrobial Resistance and Putative Virulence Genes Characterization in *Plesiomonas shigelloides* Isolated from Different Sources

**DOI:** 10.3390/antibiotics11010085

**Published:** 2022-01-11

**Authors:** Samy Selim, Mohammed S. Almuhayawi, Shadi Ahmed Zakai, Ahmed Attia Salama, Mona Warrad

**Affiliations:** 1Department of Clinical Laboratory Sciences, College of Applied Medical Sciences, Jouf University, Sakaka 72341, Saudi Arabia; 2Department of Medical Microbiology and Parasitology, Faculty of Medicine, King Abdulaziz University, Jeddah 21589, Saudi Arabia; msalmuhayawi@kau.edu.sa (M.S.A.); szakai@kau.edu.sa (S.A.Z.); aasalama@kau.edu.sa (A.A.S.); 3Department of Medical Microbiology and Immunology, Faculty of Medicine, Menoufia University, Shebin El-Kom 32511, Egypt; 4Department of Clinical Laboratory Sciences, College of Applied Medical Sciences at Al-Quriat, Jouf University, Al-Quriat 77454, Saudi Arabia

**Keywords:** shellfish, sea water, antimicrobial resistance, putative virulence genes, *Plesiomonas shigelloides*

## Abstract

*Plesiomonas shigelloides* are gram-negative, thermotolerant, motile, and pleomorphic microorganisms that are only distantly related to those of the Enterobacteriaceae and Vibrionaceae families. One of the most common sources of *P. shigelloides* contamination is human stool, but it may also be found in a wide range of other animals, plants, and aquatic habitats. Antimicrobial resistance in *P. shigelloides* from seawater and shellfish was investigated, and pathogenicity involved genes were characterized as part of this study. Out of 384 samples of shellfish, 5.7% included *P. shigelloides*. The presence of *P. shigelloides* was also discovered in 5% of the seawater sampled. The antimicrobial resistance of 23 *P. shigelloides* isolates derived from those samples was investigated. All isolates were sensitive to nalidixic acid, carbenicillin, cephalothin, erythromycin, kanamycin, tetracycline, and ciprofloxacin in the study. Several strains isolated from diseased shellfish were tested for virulence in shellfish by intraperitoneal injections. The LD_50_ values ranged from 12 × 10^8^ to 3 × 10^12^ cfu/shellfish. When looking for possible virulence factors that may play a significant role in bacterial infection in the current study, we found that all of these genes were present in these strains. These include genes such as elastase, lipase, flagellin, enterotoxin, and DNases. According to these findings, shellfish may serve as a reservoir for multi-resistant *P. shigelloides* and help spread virulence genes across the environment.

## 1. Introduction

Aquaculture’s bivalves shellfish are a major food source with a global output and commercial worth [1]. Following an increase in bivalve clam consumption, reports of infectious seafood outbreaks have become increasingly common. As bivalves grow and mature, harmful bacteria thrive and their metabolites build up in their bodies [2]. A complicated hydromechanical capacity of bivalves to filter surrounding water and accumulate harmful bacteria contained therein that may cause a variety of infectious illnesses in humans is demonstrated by their role as filter animals [3]. The consumption of raw or barely cooked bivalve shellfish also raises the danger of exposure to infectious pathogens [4]. It would be ideal to cultivate and collect mollusks in places free of coastal water contamination, but this is difficult from a production standpoint because of their scarcity of places free of contamination [5,6].

A Gram-negative, motile, oxidase-positive, and facultatively anaerobic rod, *Plesiomonas shigelloides* belongs to the Enterobacteriaceae family [7]. Most commonly found in tropical and subtropical regions, these bacteria might also be discovered in temperate and cold climates’ freshwater habitats [8]. Over the past several years, the gastrointestinal sickness associated with *P. shigelloides* has garnered growing attention, particularly in children, the elderly, and those with impaired immune systems. It has also been linked to travel-related diarrhea andpolluted water [9,10], normally from consuming raw or undercooked crabs and shellfish. For a long time, *Plesiomonas* was considered a part of the Vibrionaceae family; however, it has now been reclassified as an *Enterobacteriaceae* member [11]. *Aeromonas* and *Plesiomonas* are both members of the Vibrionaceae family, which typically inhabits the aquatic environment [12]. Freshwater and estuary habitats are home to the lone known species of *P. shigelloides* [13]. When it enters the body by water or food, it can produce mild watery diarrhea to more severe dysentery-like diarrhea, which can be acute or chronic and vary from mild to moderate to severe [14]. *P. shigelloides* has several characteristics with other pathogenic vibrios, such as disease symptoms and enrichment medium [11]. Many virulence factors have been linked to the pathogenesis of this microbe, including β-hemolysins, enterotoxins, cholera-like toxins, and probable endotoxins [15].

Many features of *Plesiomonas* molecular pathogenicity remain unclear, despite the fact that the pathogenicity of *Plesiomonas* has been studied. Unfortunately, there are no animal models available to definitively identify the virulence factors, and the one human volunteer research study that was conducted failed. The pathogenicity of *P. shigelloides* has not been fully explained, according to current findings, and more than one virulence factor is needed to induce diarrhea. None of the many suspected *Plesiomonas*-associated infection virulence factors have been universally regarded as important or frequently tested [13]. Several strains of *Plesiomonas* obtained from various sources, as well as strains of closely related bacteria, such as *Vibrio*, *Aeromonas*, or *Escherichia coli*, have been used to test the selectivity of the PCR tests for *P. shigelloides* [10]. According to Nawaz et al. [15], the hugA and 23S rDNA primers arose a non-specific amplification using an *Aeromonas* and *Plesiomonas* strains carrying the *stx2* gene; however, this is unlikely to occur in regular testing [14,16]. The discovery of appropriate targets holds great promise for improving *P. shigelloides* molecular detection. However, the improvement of detection, identification, and characterization of this bacterium necessitates more molecular inquiry. The discovery of pathogenic pathways and the understanding of *P. shigelloides* involvement in human disease are still needed. As a result, we decided to evaluate representative Plesiomonas isolates recovered from water and shellfish for their public health significance and investigate whether these isolates contained putative virulence factors associated with antimicrobial resistance. Furthermore, we studied the pathogenicity of *P. shigelloides*, which was validated by conducting infectivity testing.

## 2. Results

There were an overall 23 *P. shigelloides* recovered, including 22 from shellfish (*n* = 384) and one from water samples (*n* = 20) (Table 1). *P. shigelloides* isolates from shellfish samples totaled 5.7%. It was determined which antimicrobial drugs were most effective against the 23 *P. shigelloides* strains that had been identified so far. All of the strains tested showed some level of susceptibility to the antimicrobial drugs tested (nalidixic acid, carbenicillin, cephalothin, erythromycin, kanamycin, tetracycline, and ciprofloxacin). Table 1 lists the antibiotic resistance profiles of 23 isolates of *P. shigelloides*. Kanamycin and tetracycline were the most effective antibiotic against *P. shigelloides* (87%). Nalidixic acid had the highest level of resistance among all isolates tested (26.1%).

Figure 1 displays the frequency of the virulence factors found in the 23 *P. shigelloides* isolates. 52.2% of the isolates included the aerolysin gene, while DNases were the least common of the enzymes to have genes (34.8%). One isolate included the aerolysin/elastase/lipase/flagellin/enterotoxin/DNases mixture. Foodborne pathogens such as *P. shigelloides* are those that possess the aerolysin and haemolysin genes.

Table 2 shows the virulence assay findings of *P. shigelloides* strains. All of the examined strains were shown to be harmful to shellfish. The LD_50_ varied from 12 × 10^8^ to 3 × 10^12^ cfu/shellfish, according to the results. Haemorrhagic fins and ulcers were seen on infected mussels in the laboratory, much as they are seen in real-life epidemic cases. First-day changes remained until the seventh day of the challenge, whereas no change was found in the control shellfish. Strains of bacteria that had been introduced into dead shellfish may be recovered and isolated.

## 3. Discussion

Most *P. shigelloides* infections are disseminated by contact with polluted water [17]. To make matters worse, antimicrobial medicines are used indiscriminately on farms and in aquaculture in particular. This leads to the creation of antibiotic-resistant bacterium strains, which are a global public health issue [3,14]. It is possible that bacteria with intermediate susceptibility have already developed resistance mechanisms, as evidenced by the presence of resistant strains [9]. There were discrepancies between the automated Vitek 2 approach and the manual method for evaluating antibiotic susceptibility [11,18]. However, the gold standard approach for evaluating antimicrobial susceptibility is broth microdilution [19].

*P. shigelloides* was tested for virulence phenotypes and found to generate more virulence factors, as evidenced by increased hemolytic activity [20]. Primers have not been designed for the identification of virulence genes by polymerase chain reaction in Porphyromonas shigelloides; instead, only phenotypic studies have been conducted [21]. In order to screen for virulence genes in *P. shigelloides*, the same primers used to detect them in *Aeromonas* were employed [13]. However, none of the *PCR* products was compatible with the size expected for these genes, confirming the identification made by the Vitek 2 system and the specificity of these primers to the *Aeromonas* genus [22].

Pure cultures of *P. shigelloides* were recovered from the infected shellfish’s internal organs. All of the isolates were pathogenic for shellfish when tested. The investigated strains meet the criteria of Kubelová et al. [23] for virulence. We found that the LD_50_ values published for this fish disease by other researchers are consistent with our findings Zorrilla et al. [24]. Specific tests for shellfish pathogenicity are thus required to prove or disprove this species’ virulence [25].

Finally, shellfish and the habitat they live in are breeding grounds for germs that may be spread through water and food [26]. As a result, the presence of these bacteria in shellfish and their habitats should be continually checked, and infection control measures should be established to reduce the zoonotic risk to humans who handle animals [27].

*P. shigelloides* strains isolated from shellfish and water samples were characterized in the current study. As a result of these findings, antimicrobial and vaccination campaigns against these harmful bacteria should be taken into consideration, Because the diverse bacteria that we recovered in this study showed an overall rise in resistance to various antimicrobials [28]. One finding from this study is that the pathogenic potential of *P. shigelloides* has been broadly disseminated across isolates [29]. It was discovered that aerolysin was the most prevalent virulence factor in the isolates tested for this study. Pathogens in farmed shellfish have never been described until now [30]. Our findings imply that this bacterium species should be classified as an opportunist shellfish pathogen and included in future treatments and prophylactic for farmed shellfish. Clams and oysters, on the other hand, are known to be reservoirs for *P. shigelloides*, which has been found to have several genes for virulence and antibiotic resistance [31]. As a result, the threat to human health from eating undercooked shellfish containing *P. shigelloides* should not be disregarded. While proper heating should remove harmful germs, undercooking or cross-contamination during preparation might be a problem [32].

## 4. Materials and Methods

### 4.1. Conditions for the Isolation of P. shigelloides from Shellfish and Water Samples

Shellfish (Oyster) was acquired in Saudi Arabia’s Aljouf province from fish markets. Shellfish were kept in sterile water at room temperature. Sterile glass bottles were used to collect 1000 mL of water from 20 cm below the water’s surface and transported in one hour to the lab for bacteriological investigation. 225 mL alkaline peptone water was added to 25 g of shellfish flesh and homogenized in Stomacher Bags for 2 min aseptically (CM1028, Oxoid UK). Tryptic Soy Broth (TSB) containing 20% glycerol was injected with an aliquot of the enrichment after 18 h of incubation at 37 °C and incubated at 28 °C for 24 h [2,7,8]. A total of 23 possible *P. shigelloides* colonies were chosen for additional examination and characterization. All isolates were kept at 70 °C in 20% glycerol Luria-Bertani broth (Sigma-Aldrich, Darmstadt, Germany).

### 4.2. Phenotypic Identification

All cultures were cultivated for 18 h at 37 °C on tryptose soya agar (TSA) (CM0131, Oxoid, UK) prior to testing. Biochemical assays were used to identify *P. shigelloides* isolates, including the indole test, gas from glucose test, arginine dihydrolase, lysine decarboxylase, and ornithine decarboxylase using the Moeller’s technique, as well as the esculin hydrolysis, Voges–Proskauer, acid production from L-arabinose, lactose, sucrose, salicin, and m-inositol [11,13].

### 4.3. Molecular Identification

All strains were re-identified using 16S rDNA of obtaining restriction fragment length polymorphism patterns (RFLP) [5].

### 4.4. Antimicrobial Susceptibility Testing

Confirmed isolates of *P. shigelloides* were tested for their antibiotic susceptibility using the disc diffusion technique and the European Committee on Antimicrobial Susceptibility Testing Guidelines (EUCAST, 2018a). Tests were conducted on a set of seven antibiotics often prescribed for the treatment of gastroenteritis and infections beyond the gastrointestinal tract. As a result of the interpretation table, all of the isolates were categorized as resistant, intermediate, or susceptible (CLSI, 2018; EUCAST, 2018a, 2018b). Antimicrobial susceptibility of 23 *P. shigelloides* strains was assessed using the Kirby-Bauer disc-diffusion method [7].

### 4.5. Genetic Detection of Virulence Genes

#### 4.5.1. Nucleic Acid Isolation

Bacterial lysis was used to isolate nucleic acids, which were then extracted with phenol-chloroform and lysed using 0.5% sodium dodecyl sulfate and 5 mg of lysozyme per milliliter. To precipitate the nucleic acids, we employed ethanol as a solvent, and we prepared the solution using a Tris-EDTA buffer. It was possible to determine the amount of nucleic acid by observing the OD at 260 nm, which was then converted into a final concentration of 200 g/mL in TE buffer. In order to make template DNA for the polymerase chain reaction (PCR), the DNA was diluted to 2 g/mL in distillate H_2_O [5].

#### 4.5.2. Detection of Virulence Genes

To confirm the identification of *P. shigelloides*, a thin layer of the bacteria was spread on nutrient agar plates and incubated at 36 °C for 24 h. Suspension was created by combining nuclease-free water, vortexed, and heated at 100 °C for 10 min before being boiled to eradicate bacteria. This was followed by a 5-min centrifugation at 12,000 rpm, and the supernatant was immediately transferred to a clean 2 mL Eppendorf tube and placed on ice for template DNA extraction. This is what we used to amplify our virulence genes. The primers we used can be found in Table 3. These primers were selected because previous research had shown that they were highly specific. The total volume of PCRs was 25 μL. Table 3 provides the PCR settings needed to find the virulence genes [4,12,15]. Ethidium bromide staining (E8751- Sigma-Aldrich, USA) and 100 V were used in the following electrophoresis. A 1% (*w*/*v*) agarose gel was used in a 0.5 X TBE buffer for 45 min with a voltage of 100 V. The gene DNA ladder utilized a 100 bp molecular weight marker. An image of the gel was obtained after electrophoresis by means of the Molecular Imager Gel Doc^TM^ XR^+^ System and Image Lab^TM^ Software 170-8195 (BioRad, Hercules, CA, USA).

### 4.6. Infectivity Test

The pathogenicity of *P. shigelloides* was confirmed by conducting experimental challenges on healthy shellfish (mean weight: 2 g). Intraperitoneal injections of six shellfish per dosage were used for the 50% lethal dose (LD_50_) testing. Tests on the pathogenicity of five strains (the molecular kinds that predominated) for shellfish were conducted on these five. We centrifuged and resuspended overnight bacterial cultures the next day (PBS, pH 7). The culture (103 to 108 cfu/mL) was injected into the shellfish ventral area in 0.1 mL of consecutive 10-fold dilutions. Only PBS was used to infect the control mussels. The injected bacteria was ascribed to the deaths that occurred every day for seven days after the bacterium was collected in pure cultures from the shellfish internal organs. The LD_50_ was determined using Reed and Müench’s highest number approach [33]. The challenging experiment was performed in 35% aquaria (six shellfishper aquarium) filled with 10 L of sterilized/filtered seawater maintained at 35 (g/L or ‰) and 22 °C.

The pathogenicity of *P. shigelloides* was confirmed by conducting experiments on healthy shellfish (mean weight: 2 g). Six shellfish were injected intraperitoneally with the 50% lethal dose (LD_50_) test solution. The pathogenicity of five strains (most prevalent) for shellfish was examined. Bacterial strains that had been incubated overnight were centrifuged and resuspended in PBS (pH 7). The culture was serially diluted 10-fold before being injected into the shellfish at concentrations ranging from 10^3^ to 10^8^ cfu/mL. The control shellfish were inoculated with PBS only. Mortalities were recorded daily for seven days, and were attributed to the inoculated bacterium when that bacterium was recovered in pure cultures from the internal organs. The LD_50_ was calculated with the highest numbers method of Reed and Müench [33]. The challenging experiment was performed in 35% aquaria (6 shellfish per aquarium) filled with 10 L of sterilized/filtered seawater maintained at 35 (g/L or %) and 22 °C.

### 4.7. Statistical Analyses

Version 17.0 of the Statistical Package for Social Sciences was used for the statistical analyses (SPSS, Inc., Chicago, IL, USA). The x2-test or Fisher’s exact test was used to evaluate the statistical significance of the differences between the groups. It was determined that *p* ≤ 0.05 was statistically significant for all of the studies that were conducted.

## Figures and Tables

**Figure 1 antibiotics-11-00085-f001:**
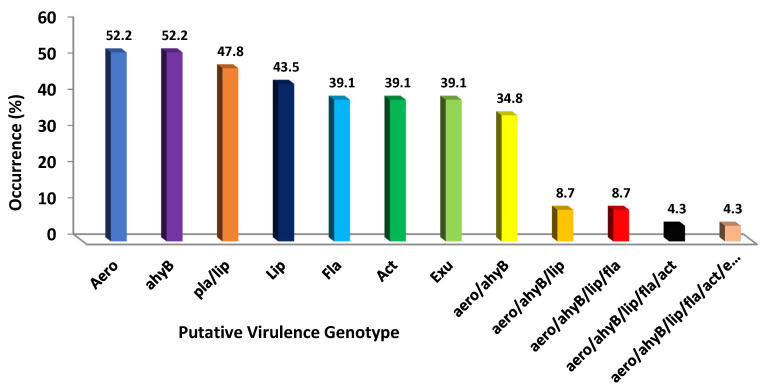
Putative virulence genotypic patterns of 23 isolates of *P. shigelloides.* Values are means of three biological replicates.

**Table 1 antibiotics-11-00085-t001:** Comparison of drug resistance patterns of 23 isolates of *P. shigelloides* isolates against antimicrobial agents.

Antimicrobial Agent	Susceptible *	Resistance
Nalidixic acid	6 (26.1%) ± 0.015	17 (73.9%) ± 1.18
Cephalothin	12 (52.3%) ± 0.23	11(47.8%) ± 0.96
Ciprofloxacin	17 (73.9%) ± 0.007	6 (26.1%) ± 0.79
Carbenicillin	11 (47.8%) ± 1.069	12 (52.3%) ± 0.009
Erythromycin	12 (52.3%) ± 1.30	11(47.8%) ± 0.12
Kanamycin	20 (87%) ± 0.99	3 (13%) ± 0.19
Tetracycline	20 (87%) ± 1.22	3 (13%) ± 2.36

* Values are means of three biological replicates.

**Table 2 antibiotics-11-00085-t002:** Test findings for the LD_50_ (cfu/shellfish) infectivity rates of five *P. shigelloides* genotypic strains after intraperitoneal inoculation.

Genotypes	LD_50_ (cfu/shellfish) ^#^
Control *	1 × 10^6^ ± 0.22
Aerolysin/Elastase	12 × 10^8^ ± 1.34
Aerolysin/Elastase/Hidrolipase	12 × 10^9^ ± 0.19
Aerolysin/Elastase/Hidrolipase/Flagellin	1 × 10^10^ ± 3.02
Aerolysin/Elastase/Hidrolipase/Flagellin/Enterotoxin	2 × 10^12^ ± 2.52
Aerolysin/Elastase/Hidrolipase/Flagellin/Enterotoxin/DNases	3 × 10^12^ ± 0.77

* Injected by phosphate-buffered saline (PBS, pH 7) only. **^#^** Values are means of three biological replicates.

**Table 3 antibiotics-11-00085-t003:** Primer sets for the detection of virulence genes in *P. shigelloides*.

Target Gene	Primer	Sequence (3ʹ-5ʹ)	Size (bp)	Reference
Elastase gene	ahyB-F	ACACGGTCAAGGAGATCAAC	540	[4]
	ahyB-R	CGCTGGTGTTGGCCAGCAGG		
Lipase gene	pla/lip-F	ATCTTCTCCGACTGGTTCGG	383–389	[4]
	pla/lip-R	CCGTGCCAGGACTGGGTCTT		
Hidrolipase gene	lip-F	AACCTGGTTCCGCTCAAGCCGTTG	65	[4]
	lip-R	TTGCTCGCCTCGGCCCAGCAGCT		
Flagellin gene	fla-F	TCCAACCGTYTGACCTC	608	[12]
	fla-R	GMYTGGTTGCGRATGGT		
Enterotoxin gene	act-F	AGAAGGTGACCACCACCAAGAACA	232	[12]
	act-R	AACTGACATCGGCCTTGAACTC		
DNases gene	exu-F	AGACATGCACAACCTCTTCC	323	[12]
	exu-R	GATTGGTATTGCCCTGCAAC		
Aerolysin gene	aeroF	TGGTAGCTAATAACTGCCAG	1170	[15]
	aeroR	GGCTTCTCTCGTTTGGCGT		

## Data Availability

The data presented in this study are available upon request from the corresponding author.

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
