# Peer review of "Distinction between Antimicrobial Resistance and Putative Virulence Genes Characterization in Plesiomonas shigelloides Isolated from Different Sources"

_antibiotics, 2022, doi:10.3390/antibiotics11010085_

Round 1

Reviewer 1 Report

The authors conducted a study on antimicrobial resistance and putative virulence genes in Plesiomonas shigelloides isolated from shellfish and water samples. The topic is interesting, the manuscript as a whole is well-organized, well-written, and updated. I have some suggestion to improve the readership of manuscript.

Title: "Isolated from several sources" was indicated by the authors, but only "Shellfish and water samples were used to isolate P. shigelloides." It would be better to include the name of the sources in the title for better readership.

Table 1-3: There is no statistical representation in the Tables. Why?

Line 157-158: Why “Shellfish were kept in sterile water at 28oC”?

Line 162: Use full form of “TSB” for the first time.

Author Response

Author's Reply to the Review Report (Reviewer 1)

Comments and Suggestions for Authors

The authors conducted a study on antimicrobial resistance and putative virulence genes in Plesiomonas shigelloides isolated from shellfish and water samples. The topic is interesting, the manuscript as a whole is well-organized, well-written, and updated. I have some suggestion to improve the readership of manuscript.

Title: "Isolated from several sources" was indicated by the authors, but only "Shellfish and water samples were used to isolate P. shigelloides." It would be better to include the name of the sources in the title for better readership.

Response: Thanks, you for your suggestion, the point that this paper is funded from the project and we are stick to use “different sources”. And the project regulation is to mandatory with that title on which we were approved on it. But we will put this in consideration in our next projects.

Table 1-3: There is no statistical representation in the Tables. Why?

Response: Thank you sir, statistical analysis added to the modified version.

Line 157-158: Why “Shellfish were kept in sterile water at 28oC”?

Response: thank you for your note, we mean in the room temperature. It has been fixed in the modified version.

Line 162: Use full form of “TSB” for the first time.

Response: thank you for your note, the full name are added in modified version

Reviewer 2 Report

After reading this manuscript I recommend to accept it in current form. Authors present interesting and well visualised results that are very valuable.

Author Response

Author's Reply to the Review Report (Reviewer 2)

Comments and Suggestions for Authors

After reading this manuscript I recommend to accept it in current form. Authors present interesting and well visualized results that are very valuable.

Response: Dear respectful reviewer, Thank you very much

Reviewer 3 Report

The manuscript Distinction between antimicrobial resistance and putative virulence genes in Plesiomonas shigelloides isolated from different sources investigates different strains of pathogenic bacteria regarding their resistance to antimicrobial substances, occurrence of genes of virulence and their lethality to shellfish.

The manuscript has interest for food safety and for the fishery sector, since filter feeding shellfish may accumulate this pathogens and lead to food-borne diseases. The results obtained in this study for Plesiomonas shigelloides pathogenic strains and its virulence contribute to the better understanding of strain variability and to the development of new tools for bacterial detection and mitigation. However, the manuscript needs further revision. An important flaw is the absence of replicates in the challenging experiment. The material and method section needs to be more clearly explained. The whole manuscript, particularly the discussion section needs to be supported with more references. The result section could be more explicative by showing plots to better visualize the outputs. In the overall, the manuscript is well written but there are many mistakes, such as missing words, duplicated names of the tables, lack of homogenization in abbreviations and so on. Bellow, a detailed list of aspects to be improved and amendments needed.    

Abstract

Line 20. Seems that something is missing at the beginning of this phrase: ”X samples out of 384…?”

Introduction

Line 42-43. Do you mean the scarcity of places free of contamination? Then, free of contamination and suitable for bivalve culture/harvesting (coastal waters).

Line 58. Delete “when it comes to habitats”

Line 62-69. This paragraph needs references.

Line 69-71. Add references for the other bacteria you mention, not only Aeromonas

Line 75-83. This is a suggestion for authors and just a matter of style. Since you finished the previous phrase speaking about detection techniques you could start the next one with the same topic, just to better link both sentences. For example:

The discovery of appropriate targets holds great promise for improving P. shigelloides molecular detection. However, the improvement of detection, identification, and characterization of this bacterium necessitate more molecular inquiry. The discovery of pathogenic pathways and the understanding of P. shigelloides involvement in human disease are still needed. As a result, we decided to evaluate representative Plesiomonas isolates recovered from water and shellfish for their public health significance and investigate whether these isolates contained putative virulence factors associated with antimicrobial resistance. Furthermore, we studied the pathogenicity of P. shigelloides, which was validated by conducting infectivity testing.

Results

Line 105. “Haemorrhagic fins and ulcers were seen on infected mussels in the laboratory” Haemorrhagic fins in mussels? In material and methods you mention shrimps. What species did you use? Please refer always to the species you used and not generalize to “shellfish” because the results you present here is for a determinate species, right?

Line 109. were

Line 111. Table 3. Intravenous or intraperitoneal? You mention intraperitoneal in Material and Methods section, please correct.

Line 113. Discussion. The whole section must be revised. Please add references (with the journal format), compare results from other papers with yours and follow a logical structure, like in the result section: occurrence in water and shellfish (in your area of study and other areas), resistance, virulence genotypic patterns, strains pathogenic to shellfish, implications for shellfish (firstly for the species you tested and then for other marine species) and for human health. Also, avoid contractions, check abbreviations, italics in species, etc. Some examples:

Line 119. Add a reference

Line 121. Add a reference

Line 123. Add a reference

Line 124. Reference, PCR and italics

Line 129. Reference

Line 130. Italics

Line 131. “That's why this species' LD50 was determined” Delete this phrase.

Line 132. References must be in the format required by this journal (with numbers).

Material and Methods

Line 157. What species and why did you choose them?

Line 167-171. Add a reference for each method.

Line 177 and 183. P. shigelloides in italics.

Line 202. This should be Table 4. Correct also in the table. The second Table 3 should be Table 4.

Line 203. Delete in Table 1 after “prior research”

Line 213. The whole section Infectivity test needs clarity. Please rewrite. For example:

In line 2015 Intraperitoneal injections of five bacterial strains were performed at six different concentration (0-X cfu/mL) in one shrimp per dosage (replicates?). 

In line 220. “Shelfish bass”. May be “ventral area”

In line 221. “At 35 percent salinity and 22 C, aquaria were used for the experimental

Infections”. This phrase can be mentioned at the beginning or at the end of the section as “The challenging experiment was performed in X aquaria (X shrimp per aquarium) filled with X liters of sterilized/filtered? seawater maintained at 35 (g/L or ‰) and 22 ºC.

In Line 222-223. This phrase is not clear to me. “…after the bacteria was collected in pure cultures from the patient’s internal organs”? or after inoculation in shrimps? Who is the patient?

Line 224. Add the reference.

Also, mention the species you used and why you chose this one in particular for your experiment.

References: Check italics in certain titles (i.e. 8, 9, 13, 14…), be consistent with journal abbreviations (i.e.  1, 2…)

Throughout the document:

Be consistent with abbreviations of percentage or % , mg or milligrams, etc. Use the short form in such cases.

Author Response

Author's Reply to the Review Report (Reviewer 3)

The manuscript Distinction between antimicrobial resistance and putative virulence genes in Plesiomonas shigelloides isolated from different sources investigates different strains of pathogenic bacteria regarding their resistance to antimicrobial substances, occurrence of genes of virulence and their lethality to shellfish.

The manuscript has interest for food safety and for the fishery sector, since filter feeding shellfish may accumulate this pathogens and lead to food-borne diseases. The results obtained in this study for Plesiomonas shigelloides pathogenic strains and its virulence contribute to the better understanding of strain variability and to the development of new tools for bacterial detection and mitigation. However, the manuscript needs further revision. An important flaw is the absence of replicates in the challenging experiment. The material and method section needs to be more clearly explained. The whole manuscript, particularly the discussion section needs to be supported with more references. The result section could be more explicative by showing plots to better visualize the outputs. In the overall, the manuscript is well written but there are many mistakes, such as missing words, duplicated names of the tables, lack of homogenization in abbreviations and so on. Bellow, a detailed list of aspects to be improved and amendments needed.

Response: Dear respectful reviewer, Thanks for positive feedback

Abstract

Line 20. Seems that something is missing at the beginning of this phrase: ”X samples out of 384…?”

Response: thank you for your note, it has been fixed in the modified version.

Introduction

Line 42-43. Do you mean the scarcity of places free of contamination? Then, free of contamination and suitable for bivalve culture/harvesting (coastal waters).

Response: Thank you for your comment. Yes, you are right we miss in expression, it has been fixed in the modified version

Line 58. Delete “when it comes to habitats”

Response: thank you for your note, it has been fixed in the modified version

Line 62-69. This paragraph needs references.

Response: thank you for your note, it has been fixed in the modified version and reference are added.

Line 69-71. Add references for the other bacteria you mention, not only Aeromonas

Response: thank you for your note, it has been fixed in the modified version References added [14-17]

Line 75-83. This is a suggestion for authors and just a matter of style. Since you finished the previous phrase speaking about detection techniques you could start the next one with the same topic, just to better link both sentences. For example:

The discovery of appropriate targets holds great promise for improving P. shigelloides molecular detection. However, the improvement of detection, identification, and characterization of this bacterium necessitate more molecular inquiry. The discovery of pathogenic pathways and the understanding of P. shigelloides involvement in human disease are still needed. As a result, we decided to evaluate representative Plesiomonas isolates recovered from water and shellfish for their public health significance and investigate whether these isolates contained putative virulence factors associated with antimicrobial resistance. Furthermore, we studied the pathogenicity of P. shigelloides, which was validated by conducting infectivity testing.

 Response: Thank you for your comment. Yes, you are right, it has been fixed in the modified version

Results

Line 105. “Haemorrhagic fins and ulcers were seen on infected mussels in the laboratory” Haemorrhagic fins in mussels? In material and methods you mention shrimps. What species did you use? Please refer always to the species you used and not generalize to “shellfish” because the results you present here is for a determinate species, right?

Response: Thank you for your comment. You are right, word shrimp wrongÙˆ it has been fixed in the modified version

Line 109. Were

Response: thank you for your note, it has been fixed in the modified version

Line 111. Table 3. Intravenous or intraperitoneal? You mention intraperitoneal in Material and Methods section, please correct.

Response: thank you for your note, it has been fixed in the modified version

Line 113. Discussion. The whole section must be revised. Please add references (with the journal format), compare results from other papers with yours and follow a logical structure, like in the result section: occurrence in water and shellfish (in your area of study and other areas), resistance, virulence genotypic patterns, strains pathogenic to shellfish, implications for shellfish (firstly for the species you tested and then for other marine species) and for human health. Also, avoid contractions, check abbreviations, italics in species, etc. Some examples:

Line 119. Add a reference

Response: thank you for your note, it has been fixed in the modified version

Line 121. Add a reference

Response: thank you for your note, it has been fixed in the modified version

Line 123. Add a reference

Response: thank you for your note, it has been fixed in the modified version

Line 124. Reference, PCR and italics

Response: thank you for your note, it has been fixed in the modified version

Line 129. Reference

Response: thank you for your note, it has been fixed in the modified version

Line 130. Italics

Response: thank you for your note, it has been fixed in the modified version

Line 131. “That's why this species' LD50 was determined” Delete this phrase.

Response: thank you for your note, it has been fixed in the modified version

Line 132. References must be in the format required by this journal (with numbers).

Response: thank you for your note, it has been fixed in the modified version

Material and Methods

Line 157. What species and why did you choose them?

Response: thank you for your note, it is Ooyster

Line 167-171. Add a reference for each method.

Response: thank you for your note, it has been fixed in the modified version

Line 177 and 183. P. shigelloides in italics.

Response: thank you for your note, it has been fixed in the modified version

Line 202. This should be Table 4. Correct also in the table. The second Table 3 should be Table 4.

Response: thank you for your note, it has been fixed in the modified version

Line 203. Delete in Table 1 after “prior research”

Response: thank you for your note, it has been fixed in the modified version

Line 213. The whole section Infectivity test needs clarity. Please rewrite. For example:

Response: thank you for your note, it was wholly rewrite.

In line 2015 Intraperitoneal injections of five bacterial strains were performed at six different concentration (0-X cfu/mL) in one shrimp per dosage (replicates?).

Response: thank you for your note, it has been fixed in the modified version - corrected and added 

In line 220. “Shelfish bass”. May be “ventral area”

Response: thank you for your note, it has been fixed in the modified version

In line 221. “At 35 percent salinity and 22 C, aquaria were used for the experimental

Response: thank you for your note, it has been fixed in the modified version

Infections”. This phrase can be mentioned at the beginning or at the end of the section as “The challenging experiment was performed in X aquaria (X shrimp per aquarium) filled with X liters of sterilized/filtered? seawater maintained at 35 (g/L or ‰) and 22 ºC.

Response: thank you for your note, it has been fixed in the modified version

In Line 222-223. This phrase is not clear to me. “…after the bacteria was collected in pure cultures from the patient’s internal organs”? or after inoculation in shrimps? Who is the patient?

Response: thank you for your note, it has been fixed in the modified version

Line 224. Add the reference.

Response: thank you for your note, it has been fixed in the modified version

Also, mention the species you used and why you chose this one in particular for your experiment.

References: Check italics in certain titles (i.e. 8, 9, 13, 14…), be consistent with journal abbreviations (i.e.  1, 2…)

Response: thank you for your note, it has been fixed in the modified version

Throughout the document:

Be consistent with abbreviations of percentage or % , mg or milligrams, etc. Use the short form in such cases.

Response: thank you for your note, it has been fixed in the modified version, change in whole text.

Round 2

Reviewer 3 Report

The manuscript has been significantly improved and several points were clarified. I will accept this new version of the manuscript.